# Low-Cost High-Resolution Potentiostat for Electrochemical Detection of Nucleic Acids and Biomolecular Interactions

**DOI:** 10.3390/mi13101610

**Published:** 2022-09-27

**Authors:** Alper Demirhan, Ece Eksin, Yalin Kilic, Arzum Erdem

**Affiliations:** 1Department of Biotechnology, Graduate School of Natural and Applied Sciences, Ege University, Izmir 35100, Turkey; 2Department of Analytical Chemistry, Faculty of Pharmacy, Ege University, Izmir 35100, Turkey; 3Department of Biomedical Engineering, Izmir University of Economics, Izmir 35330, Turkey; 4Solar Biyoteknoloji Ltd. (SolarBiotec), Bayrakli, Izmir 35530, Turkey

**Keywords:** potentiostat, differential pulse voltammetry, point-of-care testing

## Abstract

A handheld USB-powered instrument developed for the electrochemical detection of nucleic acids and biomolecular interactions is presented. The proposed instrument is capable of scanning ± 2.25 V while measuring currents up to ±10 mA, with a minimum current resolution of 6.87 pA. Therefore, it is suitable for nucleic acid sensors, which have high background currents. A low-cost microcontroller with an on-chip 16-bit analog-to-digital converter, 12-bit digital-to-analog converter, and a built-in USB controller were used to miniaturize the system. The offset voltages and gain errors of the analog peripherals were calibrated to obtain a superior performance. Thus, a similar performance to those of the market-leader potentiostats was achieved, but at a fraction of their cost and size. The performance of the application of this proposed architecture was tested successfully and was found to be similar to a leading commercial device through a clinical application in the aspects of the detection of nucleic acids, such as calf thymus ssDNA and dsDNA, and their interactions with a protein (BSA) by using single-use graphite electrodes in combination with the differential pulse voltammetry technique.

## 1. Introduction

In the last few decades, various electrochemical sensors have been developed and successfully applied in various fields over the extensive surfaces of transducers and fabricating elements. There is an increasing demand for electrochemical sensors because they are rapid, sensitive, selective, easily prepared, and low-cost devices [1]. Much effort has been put into the electrochemical detection of nucleic acids for improving the sensitivity, stability, and reproducibility [2,3,4]. 

Voltammetric techniques have the benefits of easy, rapid, and sensitive measurement up to the nano level, showing a promising future in the analyses of various analytes. Differential pulse voltammetry (DPV) and square wave voltammetry (SWV) were utilized, which are sensitive and fast techniques in comparison with the other voltammetric techniques, and the extent of the overoxidation can be controlled in a facile manner. The development of new materials as electrode materials is successfully fulfilling the need of modern electrochemical technology. In this context, instead of common electrodes, such as carbon-paste electrodes, glassy-carbon electrodes, etc., disposable electrodes, including screen-printed electrodes and pencil-graphite electrodes (PGEs), have been used for voltammetric analyses in recent years. PGEs have the advantages of being inexpensive and readily available, and they have the best mechanical resistance [5]. Nowadays, the on-site measurement demand for economic, field-utilizable, easy-to-fabricate voltammetric sensors has been steadily increasing. In this context, pencil-graphite electrodes are more popular due to their good stability, easy disposability, reproducibility, and uniform quality. Therefore, PGEs are often used as voltammetric sensors in analytical chemistry for various electrochemical measurements [5,6,7,8,9,10].

Although benchtop potentiostats are suitable for laboratories, there is an increasing need for small, low-cost, and high-performance devices to take advantage of miniaturized biosensors [11]. For this aim, several designs have been proposed [12,13,14], but these works either require external peripherals (e.g., DACs and ADCs), which leads to an increase in the cost and size of the sensors, or they use the low-resolution internal peripherals of Arduino. Additionally, several analog front ends (AFEs) are proposed in the literature. Cruz et al. [15] and Yokus et al. [16] have suggested using a programmable AFE (LMP91000) (Texas Instruments, Dallas, TX, USA) that is capable of performing voltammetric and amperometric protocols. However, this AFE can produce bias voltages only between 0% and 24% of the reference voltage, and this limits the scan resolution, current resolution, and maximum scan range [17,18]. With a 5 V reference voltage, the LMP91000 can only sweep ±1.2 V with a step potential of 100 mV. The high step potential results in more capacitive current, and this limits the sensitivity of the sensor and scan rate. Another drawback is that it cannot sweep potentials between 0 mV and ± 15 mV because it requires a minimum of 1.5 V as a voltage reference. Saygili et al. proposed a solution to these problems by using a dual LMP91000 [19]. They used one AFE to bias only the working electrode (WE), and another AFE to bias only the reference electrode (RE) and counter electrode (CE). Due to their use of a DAC to bias the reference voltage, they achieved sweep potentials of ±2 V, with a resolution of less than 1 mV. However, because the WE potential in this application has to be altered to adjust the bias voltage, this architecture limits the dynamic range of the current measurement. Gu et al. have suggested an architecture similar to the one proposed in this paper [20]. They used a mixed-signal microcontroller (STM32F373) (STMicroelectronics, Geneva, Switzerland) for the waveform generation, data acquisition, and data processing. Although their sweep range was limited to ±1 V, and due to having used only one gain resistor, they could only record currents of ±100 µA. Moreover, they used a 1 kHz analog low-pass filter, which cannot eliminate the possible 50/60 Hz power-line interference and other low-frequency noise sources.

However, nucleic acid-based electrochemical sensors have high background currents, and hence, they require high-resolution measurements at high bias potentials. Therefore, the proposed designs are not able to satisfy the need for nucleic acid sensors. In this work, we have proposed a high-resolution instrument that has a high dynamic range to measure signals. The potential of the working electrode held constant, and the waveform generator output was amplified to apply a ±2.25 V sweep range. A switching network with eight gain resistors was used to obtain a more dynamic range for the current measurement, for which measuring currents up to ±10 mA was possible. The minimum current resolution reached 6.87 pA. Utilizing the internal ADC and DAC of the microcontroller (MCU) allowed us to show a calibration procedure and obtain similar responses when compared with external ADC and DAC devices. A digital low-pass filter, in addition to analog filters, was used to remove any noise. Therefore, a programmable and very sharp filtering response for frequencies lower than 10 Hz was obtained. 

Scientists first microscopically discovered the association of proteins with nucleic acids. Then, in vitro and in vivo assays demonstrated that proteins interact with DNA and RNA and effect the structure and function of nucleic acid [21]. Consequently, it was found that protein–DNA interactions play a key role in many biological processes, such as the regulation of gene expression, DNA replication, repair, transcription, and recombination, and the packaging of chromosomal DNA. In some cases, the definition of the functions of the protein–nucleic acid complex can be a key point for an understanding of some disease mechanisms [22]. In this respect, we aimed to apply our miniature device to the electrochemical monitoring of the surface-confined interaction of proteins with DNA. Albumin, as the most abundant and major protein of the blood, plays an important role in many biological processes. In fact, human serum albumin can bind several molecules, such as bases, nucleotides, RNA, and DNA [23]. With this aim, the target protein was chosen as BSA in the present study. Accordingly, its interaction with ssDNA or dsDNA at the surfaces of PGEs was explored by using the developed instrument.

To the best of our knowledge, a miniaturized handheld potentiostat has been developed and applied for the electrochemical monitoring of the biointeraction process between nucleic acid and BSA for the first time in the literature. A low-cost MCU with internal ADC, DAC, and USB controllers was used to obtain a similar performance to the commercial potentiostats, but with a lower cost, with the help of a calibration procedure and digital filtering. The performance of the developed instrument in the detection of nucleic acids (DNA, etc.) and proteins was also investigated in comparison to the Autolab PGSTAT-302, which is a widely used electrochemical analyzer, under the same conditions.

## 2. Materials and Methods

### 2.1. Instrument Architecture

The architecture of the hardware is shown in Figure 1. The STM32F373 is a mixed-signal MCU that has a rich set of peripherals, such as an ADC, DAC, and USB, which help to minimize the circuit and lower the cost. The built-in USB controller is used in full-speed mode to establish the communication protocol, instead of a UART–USB converter, as in Gu et al. [20].

The cost of the components required to enable our instrument was less than 40 US Dollars (USD), as shown in Table 1.

### 2.2. Power System

The bus voltage of a USB port is used to power the instrument. Although USB ports can provide voltage up to 5.25 V, this value highly depends on the total current consumption of the USB port. Therefore, other devices connected to the same USB hub may affect the output voltage. A 4.5 V voltage reference (REF5045) (Texas Instruments, Dallas, TX, USA) is used to power op-amps to provide the maximum available voltage. To prevent high-speed switching noises caused by the digital circuitry of the MCU, separate power supplies for the analog and digital divisions of the MCU are used. The analog circuitry of the MCU is powered with a 3.0 V reference voltage (REF5030) (Texas Instruments), which has a 3 ppm/°C temperature drift and 0.05% accuracy, while the digital circuitry is powered with a 3.3 V-output high-current (250 mA) MCP1703 (Microchip Technology, Chandler, AZ, USA) voltage regulator.

### 2.3. Signal Generation

The analog waveform is generated by the internally buffered DAC of the MCU, and it has a 4 µs rise time and 12-bit resolution, which corresponds to 0.732 mV per the least significant bit (LSB) with a 3.0 V voltage reference. However, to increase the dynamic range of the experiments, the A1 amplifier, shown in Figure 1, was used to multiply the DAC output by 1.5, and therefore, 1 LSB corresponds to 1.1 mV, which is a sufficient resolution for voltammetric experiments. As Gu et al. [20] suggested, this resolution can be easily well adjusted by summing it with the second DAC of the MCU in the voltage-divider configuration. Amplification by 1.5 makes the waveform output span 4.5 V. REF5045, which is the voltage reference with a 4.5 V output, was used to create the 2.25 V bias voltage for the WE, and to supply low noise and high-precision voltage for the op-amps and analog multiplexer. Therefore, the scan range can cover up to ±2.25 V, depending on the gain of the TIA and the value of the CE. Op-amp A2 acts as an error amplifier, which aims to supply the necessary current to hold the reference electrode at the desired potential. Therefore, the buffer voltage of the reference electrode provides negative feedback to A2, and the output of A2 drives the CE, depending on the error between A1 and the RE.

### 2.4. Data Acquisition

To provide an optimum range for the current measurements, the potential of the working electrode was held constant at 2.25 V. Therefore, the bias voltage on the sensor becomes ($V_{DAC}$ × 1.5) − 2.25 V. The TIA output can be written as V_out_ = 2.25 V − $I_{sensor}\times R_{gain}$. The gain of the TIA is configured with MAX4617, which is an eight-channel analog switch and has a low-leakage current (1 pA at 20 °C) and 8 Ω on-resistance. The switch-on and -off times are 10 ns and 15 ns, respectively. Therefore, by using values between 220 Ω and 10 MΩ, current values between 10.22 mA and 6.87 nA could be measured. Because the ADC has 16-bit resolution, the current resolutions with the maximum and minimum gains are 31.19 µA and 6.87 pA, respectively. The output of the TIA is divided by 1.5 to scale because the reference voltage of the ADC is 3.0 V. The internal ADC of the MCU has a 16-bit resolution up to a 50 kilo-sample per second (kSPS) rate, and it is used to measure the output of the TIA. Because the ADC has a delta–sigma architecture, there is a need for a reservoir capacitor before the ADC input, which helps to filter the switching spikes caused internally by the ADC and provides a charge to the sampling capacitor of the ADC. However, connecting it directly to the TIA output causes oscillation because the op-amps have limited capacity to drive capacitive loads, and therefore a resistor is put between a capacitor and low-pass filter, which also acts as an anti-aliasing filter with a cut-off frequency (f_c_). Although it is possible to configure the f_c_ very close to the DC, this limits the time response of the circuit for protocols such as differential pulse voltammetry because the generated waveform contains square waves. In our DPV experiments, the period of the square wave was 40 ms, which corresponds to 25 Hz, but considering its harmonics, the f_c_ must be higher than this value. Therefore, the value of the resistance was selected as 1 kΩ, and value of the capacitance was selected as 100 nF, which corresponds to a 1.6 kHz 3 dB cut-off frequency. Digital filtering is utilized in addition to analog filtering for noise reduction. A number of measurement points affect the signal smoothness. On the one hand, if high numbers of samples are taken by decreasing the step size, then smoother voltammograms can be obtained. However, in this case, a digital low-pass filter should be used to filter interferences, such as 50/60 Hz. On the other hand, if the step size is large, which means that the period is longer as well, then the signal can be averaged for at least 20 ms to filter the power-line interferences, without using more complex software techniques.

### 2.5. Calibration

To calibrate the offset voltages caused by the leakage currents and op-amps, the waveform generator output and ADC input value were measured with a 6½-digit digital multimeter (34461A, Keysight, Santa Rosa, CA, USA), while no load was connected to calculate the offset deviations from the ideal values. These values are saved in the flash memory of the MCU for the initial automatic calibration at power-on to provide the values to be subtracted from each measurement. 

### 2.6. User Interface

The user interface was written in C++ by using the QT framework for desktop computers, and it was designed by GalvanoPlot (İzmir, Turkey). A virtual COM port was created for the USB communication between the computer and MCU. An example experiment window for the CV is shown in Figure 2. For CV experiments, users enter the potential scan range, scan rate, and current resolution. For DPV experiments, users enter the scan range, step voltage, pulse period, pulse width, pulse amplitude, and current resolution.

### 2.7. Electrochemical Sensing of Nucleic Acids and Its Biointeractions

#### 2.7.1. Apparatus

DPV measurements were undertaken in a Faraday cage using both the developed instrument and the Autolab PGSTAT-302 electroanalysis system as a commercial potentiostat. All measurements were performed with a traditional three-electrode system using a disposable pencil-graphite electrode, a platinum wire, and an Ag/AgCl/KCl/3M (BAS, Model RE-5B, W. Lafayette, IN, USA) as the working, counter, and reference electrodes, respectively. The experimental setup containing an electrochemical cell and three-electrode system in combination with the developed instrument is given in Figure 3. The PGEs were prepared for electrochemical measurement according to the information given in our earlier studies [8,9].

#### 2.7.2. Chemicals

All chemicals were of analytical reagent grade and were supplied from Sigma-Aldrich and Merck. Bovine serum albumin (BSA), calf thymus single-stranded DNA (ssDNA), and calf thymus double-stranded DNA (dsDNA) were purchased from Sigma-Aldrich. Milli-Q ultrapure water was used to freshly prepare all aqueous solutions. The stock solutions of ssDNA and dsDNA were prepared in Tris-HCl buffer solution (TBS) (pH 7.00) and kept frozen. Diluted solutions of ssDNA or dsDNA were prepared in 0.50 M acetate buffer solution (ABS) containing 20 mM NaCl (pH 4.80). The stock solution of BSA was prepared in ultrapure distilled water. Diluted BSA solutions were prepared in 0.05 M phosphate buffer solution (PBS) containing 20 mM NaCl (pH 7.40).

#### 2.7.3. ssDNA or dsDNA Immobilization onto the Surface of PGEs

A DNA biosensor was developed by immobilizing DNA onto the electrode surface via passive adsorption by dipping PGEs into 100 µL of ssDNA or dsDNA solution and maintaining them over 30 min [9]. Each ssDNA/dsDNA-immobilized PGE was washed twice with ABS to eliminate nonspecific adsorptions.

#### 2.7.4. BSA and DNA Interaction on the Surface of PGEs

The interaction step involved dipping the DNA (ssDNA or dsDNA)-immobilized PGEs into the 6 µM BSA solution in 0.05 PBS (pH 7.40). Then, each electrode was washed with PBS before voltammetric measurement.

#### 2.7.5. Voltammetric Measurements

DPV measurements were performed between the potentials of 0 V and +1.40 V at a pulse amplitude of 50 mV and scan rate of 50 mV/s. For the electrochemical characterization of the electrode, CV measurements were performed in a potential range from −1 V to +1 V, with scan rates from 5 mV/s to 200 mV/s in the solution containing 2 mM K_3_Fe(CN)_6_ and 2 mM K_4_Fe(CN)_6_ in 0.1 M KCl [9]. A digital low-pass filter with a 5 Hz cut-off frequency was applied to all measurements by the developed device. Because no postfiltering was applied to the signals measured in the experiments with the Autolab PGSTAT-302, raw data were recorded for each measurement. 

## 3. Results and Discussion

Following the offset calibration, the gain error is calculated by performing cyclic voltammetry using a resistor with a known value (4.697 kΩ), and a 3.23% deviation is found from an ideal curve, as shown in Figure 4. The gain error (*GE*) is calculated by the following formula:
(1)$$GE=\frac{C_{r2}-C_{r1}}{C_{i2}-C_{i1}}$$

Therefore, the calibrated value of the *ADC* is calculated after each measurement by:
(2)$$ADC_{calibrated}=\frac{ADC_{raw}-offset}{GE}$$

Another experiment was performed with both devices in order to calculate the electroactive surface areas (A) of the PGEs, which were calculated using the Randles–Sevcik equation [24]. The corresponding anodic current (I_a_) and cathodic current (I_c_) and the electroactive surface areas of the PGEs are listed in Table 2. With the developed potentiostat, the I_a_ and I_c_ were 16.6 % and 35.1 % higher, respectively, than those measured with the commercial potentiostat.

CV studies were performed at various scan rates to demonstrate the custom system, and the results are shown in Figure 5. The relationship between the square root of the scan rate and the peak current is found to be linear for both the anodic peak current and cathodic peak current, with R^2^ = 0.99, as shown in Figure 5B. The linearity between the square root of the scan rate and peak current is an indication of a diffusion-controlled redox phenomenon [25,26].

Nucleic acid detection by electrochemical biosensors has received a great deal of interest due to the high sensitivity allowed by electrochemical techniques [10,27] in comparison with other techniques, such as the ELISA or spectroscopic techniques. The most promising application of the technology is in a handheld device for point-of-care testing. These types of biosensors provide sensitive and selective detection and the quantitation of nucleic acids. Detection is possible in clinical samples, such as whole blood, serum, and urine [1]. Therefore, in this part of our work, the applications of the designed potentiostat on the quantitative determination of ssDNA and dsDNA are presented. The performance of the potentiostat is compared with the Autolab PGSTAT-302. As seen in Figure 6C, with the increasing concentration of ssDNA, the current is proportionally increased over a wide concentration range of 2–20 µg/mL using the presented instrument. The linear regression equation was I (µA) = 0.25 [ssDNA] (µg/mL) + 0.31, with a correlation coefficient of 0.99 (Figure 7A). Conversely, when using the Autolab PGSTAT-302, there was a proportional increase observed at the current over a concentration range of ssDNA from 2 to 12 µg/mL (Figure 6A). The linear regression equation was I (µA) = 0.79 [ssDNA] (µg/mL) + 1.31, with a correlation coefficient of 0.98 (Figure 7A). The limit of detection (LOD) was estimated as explained by Miller and Miller [28], with a regression equation and the definition as “y = yB + 3SB” (yB is the signal of the blank solution, and SB is the standard deviation of the blank solution). The values of the LOD obtained by the developed instrument and Autolab PGSTAT-302 were calculated as 1.25 µg/mL and 1.65 µg/mL, respectively. Similarly, the effect of dsDNA was studied in the case of various concentration of dsDNA from 2 to 14 µg/mL (Figure 6G). The current was increased in the case of an increased dsDNA concentration, as expected. The linear regression equation was I (µA) = 0.14 [dsDNA] (µg/mL) + 0.44, with a correlation coefficient of 0.99 (Figure 7B). The response of the increased dsDNA concentration from 2 to 18 µg/mL using the Autolab PGSTAT-302 is shown in Figure 6E. The linear regression equation for the Autolab PGSTAT-302 was I (µA) = 0.31 [dsDNA] (µg/mL) + 0.81, with a correlation coefficient of 0.99 (Figure 7B). The LODs [28] of the developed instrument and Autolab PGSTAT-302 were calculated as 1.10 µg/mL and 0.96 µg/mL, respectively. Line graphs (Figure 6B,D,F,H) representing the average guanine oxidation signals related to the ssDNA or dsDNA-immobilized PGEs in the case of different DNA concentrations were used to observe the relationship between the DNA concentration and current. The current is proportionally increased in the case of an increased DNA concentration until the PGE surface is saturated. 

The quantitative evaluation of the interaction between DNA and protein is essential for many biological processes, including function and the developmental stages of many diseases. In this respect, we aimed to apply this miniature device for the electrochemical monitoring of the surface-confined interaction of protein with DNA. Under this aim, the target protein was chosen as BSA, and accordingly, its interaction with ssDNA or dsDNA at the surfaces of PGEs was explored by the developed instrument. The effect of the interaction time was first studied in different times, varying from 5 to 60 min. Before and after the interaction, both the oxidation signals of the BSA and guanine were measured at potentials of +0.79 V and +1.01 V, respectively, in the same voltammetric scale (shown in Figure 8).

The changes obtained at both signals of the BSA and guanine were calculated and are presented in Table 3 for both ssDNA and dsDNA. The oxidation signal of the BSA gradually decreased, while the interaction time increased. Similarly, the oxidation signal of the guanine gradually decreased in cases of increased interaction times. These results are consistent with earlier reports [29,30,31]. This phenomenon could be explained as more BSA interaction with ss/dsDNA occurring over longer durations. The highest decrease percentage was obtained with the surface-confined interaction performed for 60 min.

## 4. Conclusions

A miniaturized potentiostat was developed and applied for the electrochemical monitoring of the biointeraction process between nucleic acid and BSA for the first time in the literature. A low-cost MCU with internal ADC, DAC, and USB controllers was used to obtain a similar performance as commercial potentiostats, but with a lower cost, with the help of a calibration procedure and digital filtering. 

We successfully demonstrated the performance of the developed instrument by comparing it under the same conditions with the Autolab PGSTAT-302, which is a widely used electrochemical analyzer for the development of biosensors for the detection of nucleic acids (DNA, etc.) and proteins. The proposed architecture could be applied to other electrochemical protocols, such as square wave voltammetry and electrochemical impedance spectroscopy. The detection of calf thymus ssDNA and dsDNA by using disposable PGEs was also successfully demonstrated in this study. The results obtained with the Autolab PGSTAT-302 device were found to be similar to the ones measured by the developed miniaturized potentiostat. The LOD of the ssDNA was calculated based on the results obtained by the developed instrument, and it was found to be 1.25 µg/mL, whereas it was found to be 1.65 µg/mL by the Autolab PGSTAT-302. On the other hand, the LOD of the dsDNA was calculated with the developed instrument as 1.10 µg/mL, whereas it was 0.96 µg/mL by the Autolab PGSTAT-302. The sensor sensitivities were estimated for ssDNA and dsDNA with both instruments from the slope of the calibration curve divided by the surface area of a PGE. For ssDNA using the developed potentiostat and commercial device, the potentiostat sensitivities were calculated as 0.737 µA·mL/µg·cm^2^ and 2.714 µA·mL/µg·cm^2^, respectively. For dsDNA using the developed potentiostat and commercial device, the sensitivities were calculated as 0.423 µA·mL/µg·cm^2^ and 1.065 µA·mL/µg·cm^2^, respectively. The higher sensitivity was obtained by using the commercial device. Although a lower signal was obtained with the developed device, the electrochemical detection of DNA was performed in the same concentration range of DNA and resulted with a closer detection-limit values with the commercial one. The comparison of the performances of the developed potentiostat and the commercial device are given in Table 4. The developed potentiostat has a significant advantage over the commercial device, such as handheld portability, and even the sensitivity was lower. Moreover, the electrochemical monitoring of the interaction of BSA with nucleic acids was also successfully performed by using this instrument. Because BSA can bind several molecules, including DNA, we aimed to apply our miniature device for the electrochemical monitoring of the surface-confined interaction of DNA with BSA. The sensing system was sensitive enough to detect the surface-confined interaction.

Although the internal peripherals of microcontrollers have poorer specifications compared with external components, we have shown that they can perform as good as high-performance external components with a proper calibration. Therefore, the proposed potentiostat has the advantage of a simple, low-cost, and compact architecture, with a minimum number of external circuit components. For some applications, the ADC and DAC of the microcontroller could be limiting factors, but this can be overcome by using signal-processing techniques. For example, the ADC is 16-bit, but because its sample rate is high (>64 kSPS), oversampling can be used to increase its resolution. Because it has two DACs with 12-bit resolution, these DACs can be summed with an op-amp circuit to create a double-precision DAC. Based on the promising outcomes of our study, ongoing studies on the detection of different types of biointeractions (drugs, environmental pollutants, etc.) are still being investigated by using this miniature device in our lab.

## Figures and Tables

**Figure 1 micromachines-13-01610-f001:**
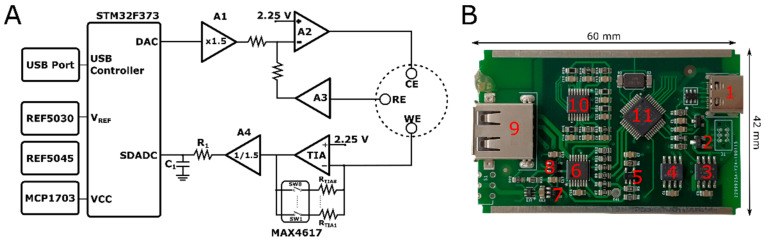
(**A**) Schematics of the hardware: STM32F373: microcontroller, ARM M4 cortex architecture, built-in ADC, DAC, and USB; MAX4617: 8:1 analog multiplexer with low current leakage; A1: operational amplifier with a gain of 1.5; A2: control amplifier to drive counter electrode; A3: buffer for reference electrode; TIA: transimpedance amplifier; A4: ADC driver with 2/3 gain; REF5045: 4.5 V reference voltage; REF5030: 3.0 V reference voltage to power analog peripherals of MCU; MCP1703: 3.3 V regulator to power digital peripherals of MCU; CE: counter electrode; RE: reference electrode; WE: working electrode. (**B**) Printed circuit board: 1: USB connector for communication; 2: 3.3 V regulator; 3: 4.5 V reference voltage; 4: 3.0 V reference voltage; 5: ADC driver; 6: op-amp; 7: op-amp; 8: op-amp; 9: USB connector for electrodes; 10: analog switch; 11: microcontroller.

**Figure 2 micromachines-13-01610-f002:**
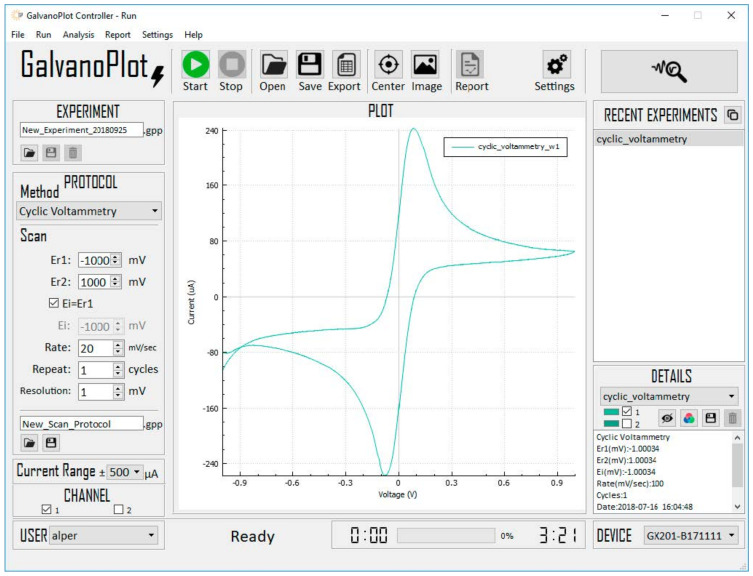
Screenshot of user interface designed in Qt framework.

**Figure 3 micromachines-13-01610-f003:**
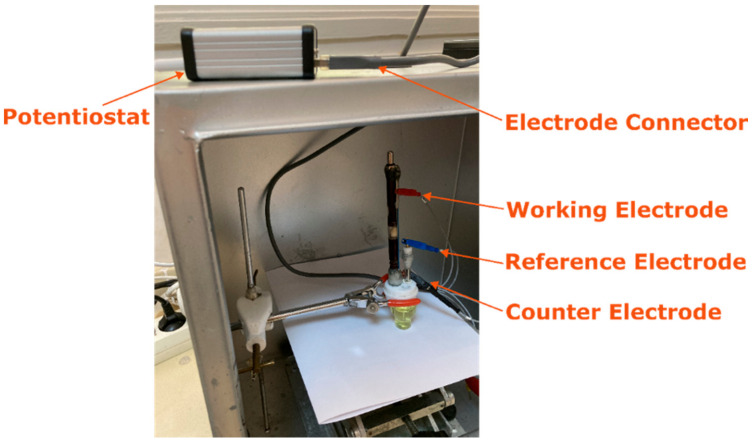
Experimental setup.

**Figure 4 micromachines-13-01610-f004:**
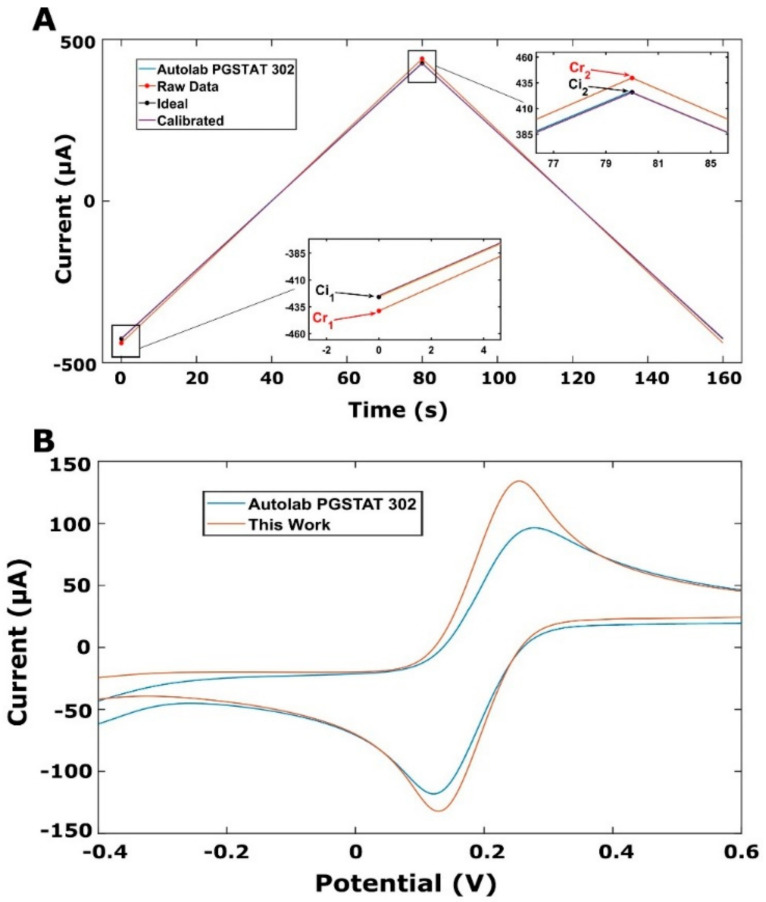
(**A**) Calibration curve of 4.697 kΩ resistance when CV signal with a potential range from −2 V to +2 V is applied. The resistor was tested both with the developed device (this work) and Autolab-PGSTAT-302. A small deviation from the ideal curve was found by the calculation of the tangent of the raw data. This value is called the gain error and it is used to calibrate the ADC after the measurement of each piece of data. Ci_1_: the lowest value of the ideal data; Ci_2_: the highest value of the ideal data; Cr_1_: the lowest value of the raw data; Cr_2_: the highest value of the ideal data. (**B**) CV characterizations of both devices.

**Figure 5 micromachines-13-01610-f005:**
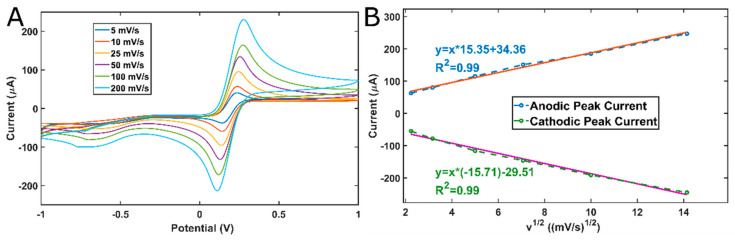
Results of CV measurements: (**A**) CV voltammograms at different scan rates; (**B**) the relationship of the peak current with the square root of the scan rate. R^2^ is 0.99 for both anodic and cathodic peak currents.

**Figure 6 micromachines-13-01610-f006:**
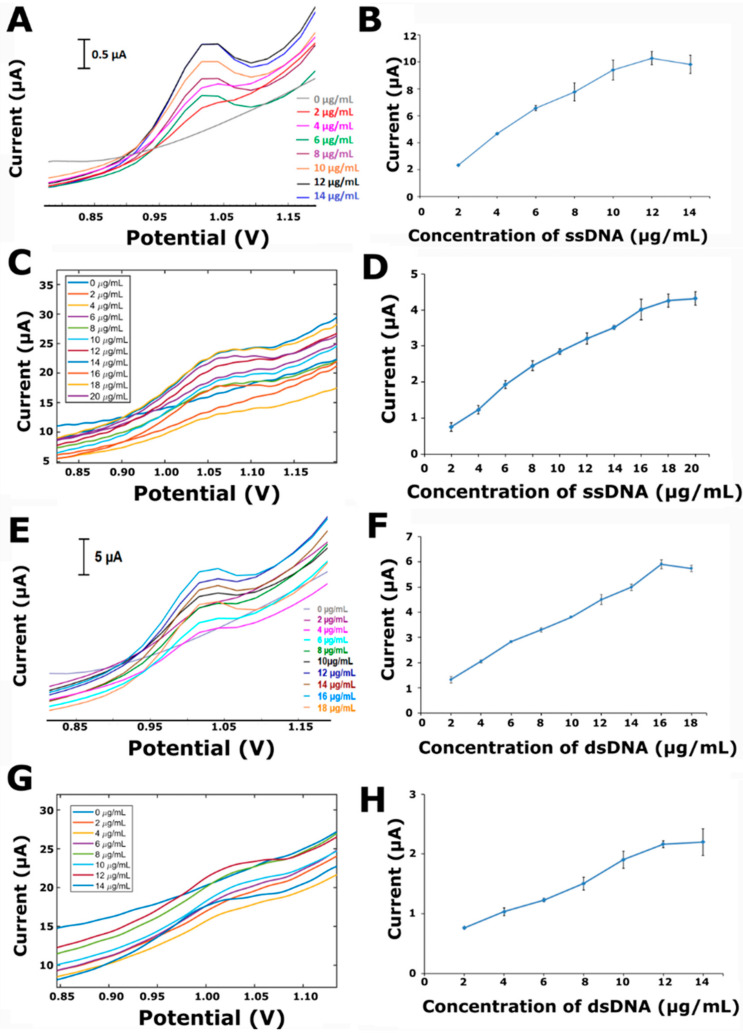
Effects of DNA concentrations on the electrochemical signal measured by the developed instrument and Autolab-PGSTAT-302. DPVs representing the guanine signals related to the different concentrations of ssDNA using (**A**) Autolab PGSTAT-302, and (**C**) the developed instrument, and the guanine signals related to the different concentrations of dsDNA using (**E**) Autolab PGSTAT-302, and (**G**) the developed instrument. (**B**) Line graph presenting average guanine signals (*n* = 3) related to ssDNA-immobilized PGEs in the case of different ssDNA concentrations from 2 to 14 µg/mL with Autolab PGSTAT-302. (**D**) Line graph representing the average guanine signals (*n* = 3) related to ssDNA-immobilized PGEs in the case of different ssDNA concentrations from 2 to 20 µg/mL with the developed instrument. (**F**) Line graph representing average guanine signals (*n* = 3) related to dsDNA-immobilized PGEs in the case of different ssDNA concentrations from 2 to 18 µg/mL with Autolab PGSTAT-302. (**H**) Line graph representing average guanine signals (*n* = 3) related to dsDNA-immobilized PGEs in the case of different dsDNA concentrations from 2 to 14 µg/mL with the developed instrument.

**Figure 7 micromachines-13-01610-f007:**
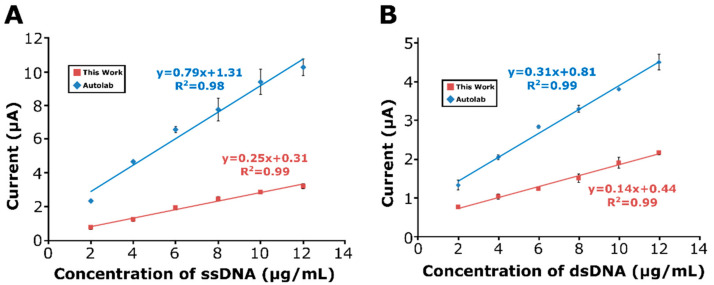
Calibration plots representing the guanine signals related to different concentrations of (**A**) ssDNA and (**B**) dsDNA using Autolab PGSTAT-302 and the developed instrument.

**Figure 8 micromachines-13-01610-f008:**
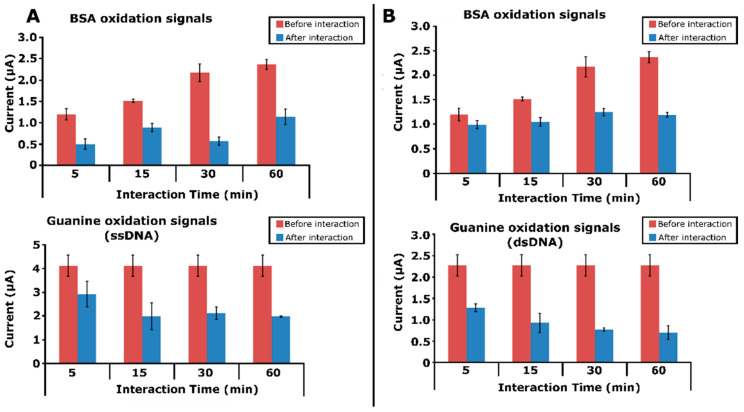
Electrochemical detection of interaction of BSA with ssDNA and dsDNA. Histograms representing the average oxidation signals of BSA and guanine, measured before and after the interaction of BSA with (**A**) ssDNA and (**B**) dsDNA in the case of different interaction times (*n* = 3).

**Table 1 micromachines-13-01610-t001:** Bill of materials.

Name	Description	Quantity	Unit Price (USD)	Total (USD)
**Passive Elements**	Resistors and Capacitors	39	0.01	0.39
**STM32F373**	MCU	1	7.94	7.94
**MAX4617**	Analog Switch	1	4.56	4.56
**AD8605**	Single Op-Amp	1	2.50	2.50
**AD8608**	Quad Op-Amp	2	6.12	12.24
**REF5030**	3.0 V Reference	1	4.49	4.49
**REF5045**	4.5 V Reference	1	4.49	4.49
**MCP1703**	3.3 V Regulator	1	0.73	0.73
**USB Connector**	USB Mini-b Connector	1	1.31	1.31
**Crystal**	16 MHz Oscillator	1	1.00	1.00
**Total Cost:**				39.65

**Table 2 micromachines-13-01610-t002:** The average values (*n* = 3) of anodic and cathodic peak currents (Ia, Ic), and electroactive surface areas (A) for PGEs, measured and calculated by commercial or developed (this work) potentiostat.

Device	I_a_ (µA)	I_c_ (µA)	A (cm^2^)
**Commercial potentiostat**	96.44 ± 1.22	98.57 ± 0.28	0.291
**Developed potentiostat**	112.44 ± 1.08	133.17 ± 3.63	0.339

**Table 3 micromachines-13-01610-t003:** The average oxidation signals of BSA and guanine (*n* = 3) measured before/after interactions (I_before_, I_after_, respectively) performed at different times with the decrease in the ratios obtained at both signals calculated after the interaction of BSA and ssDNA or dsDNA at different interaction times.

Type of DNA	Time (min)	BSA I_before_ (µA)	BSA I_after_ (µA)	Decrease % at BSA Signal	Guanine I_before_ (µA)	Guanine I_after_ (µA)	Decrease % at Guanine Signal
**ssDNA**	5	1.19 ± 0.13	0.50 ± 0.12	58%	4.11 ± 0.44	2.91 ± 0.54	29%
	15	1.51 ± 0.03	0.88 ± 0.09	41%		1.99 ± 0.56	51%
	30	2.17 ± 0.20	0.57 ± 0.09	73%		2.12 ± 0.26	48%
	60	2.36 ± 0.11	1.14 ± 0.17	52%		1.97 ± 0.03	52%
**dsDNA**	5	1.19 ± 0.13	0.98 ± 0.07	17%	2.27 ± 0.25	1.28 ± 0.08	43%
	15	1.51 ± 0.03	1.04 ± 0.08	31%		0.93 ± 0.22	38%
	30	2.17 ± 0.20	1.24 ± 0.07	42%		0.77 ± 0.04	66%
	60	2.36 ± 0.11	1.18 ± 0.04	69%		0.70 ± 0.16	69%

**Table 4 micromachines-13-01610-t004:** A comparison of the performances of the developed potentiostat and other commercial potentiostats.

Device	Electrochemical Method	Electrode Type	Analyte	LOD	Ref.
**LMP91000**	CV	Gold IDE	Cortisol	1 pM	[15]
**LMP91000**	CV, Amperometry	Carbon SPE	Cortisol	74.0 nmol L^−1^	[32]
**ADucM355**	SWV	Glassy Carbon Electrode	TNT	0.25 mg/L	[33]
**Custom** **AFE**	Chronoamperometry	Carbon SPE	HRP	0.83 ng mL^−1^	[34]
**Custom** **AFE**	CV, DPV	Carbon SPE	Cd^2+^, Pb^2+^	1 µg L^−1^ Cd^2+^, 0.5 µg L^−1^ Pb^2+^	[20]
**Autolab PGSTAT-302**	CV, DPV	PGE	ssDNA, dsDNA	1.65 µg/mL ssDNA, 0.96 µg/mL dsDNA	This work
**Current** **Prototype**	CV, DPV	PGE	ssDNA, dsDNA	1.25 µg/mL ssDNA, 1.10 µg/mL dsDNA	This work

## Data Availability

The data presented in this study are available within the article. Other data that support the findings of this study are available upon request from the corresponding authors and coauthors.

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
