# Peer review of "Low-Cost High-Resolution Potentiostat for Electrochemical Detection of Nucleic Acids and Biomolecular Interactions"

_micromachines, 2022, doi:10.3390/mi13101610_

Round 1
Reviewer 1 Report
The works presented in this paper are related to developing the low-cost potentiostat. However, the authors should add several details to explore the studies thoroughly.
- Add details of the components' price.
- The color number in Figure 1B is unclear. I think it is better to share the PCB layout file and make it open source to gain early adopter interest and improve the current prototype faster in the future.
- How about the user interface of this system? Is this using a mobile app or PC/laptop? Please clearly explain and add the flow chart on how to use the prototype. The app also can be shared as an open source.
- It is better to show the photo of the prototype during the experimental setup to gain readers' interest in the real size of the device.
- Regarding CV performance, why are the peaks of the prototype higher than the commercial? How much is the difference (in %) of peaks? Please add CV response from the commercial potentiostat.
- Figure 4 needs to resize. It is unclear and too small.
- Could you check the DPV performance of K3Fe(CN)6 and K4Fe(CN)6 solution on a varied concentration? I want to know the signal responses because it seems the DPV peaks are not sharp in the data in Figure 4A-D.
- It seems the signal smoothness on the prototype still quite rough. How to improve the signal? Add the suggestions.
- Add a comparison table for the performance of the current prototype with several commercial products and other previous research
Author Response
Micromachines
micromachines-1890500
Title:
A low-cost, high-resolution potentiostat for electrochemical detection of nucleic acids and biomolecular interactions
12th September 2022
The list of answers to the comments of reviewers
Thank you for valuable comments of Reviewer#1, Reviewer#2, and Reviewer#3. Manuscript is revised/corrected according to each comment pointed by reviewers. The revised/corrected parts in the manuscript are shown highlighted in yellow.
Reviewers' comments:
Reviewer 1
Open Review
Comments and Suggestions for Authors
The works presented in this paper are related to developing the low-cost potentiostat. However, the authors should add several details to explore the studies thoroughly.
- Add details of the components' price.
Answer: Authors would like to thank to the Reviewer #1 for their valuable comments. The bill of materials is given in Table 1 and added to the main text as below.
Name |
Description |
Quantity |
Unit price ($) |
Total ($) |
Passive elements |
Resistors and capacitors |
39 |
0.01 |
0.39 |
STM32F373 |
MCU |
1 |
7.94 |
7.94 |
MAX4617 |
Analog Switch |
1 |
4.56 |
4.56 |
AD8605 |
Single Op-amp |
1 |
2.50 |
2.50 |
AD8608 |
Quad Op-amp |
2 |
6.12 |
12.24 |
REF5030 |
3.0V Reference |
1 |
4.49 |
4.49 |
REF5045 |
4.5V Reference |
1 |
4.49 |
4.49 |
MCP1703 |
3.3V Regulator |
1 |
0.73 |
0.73 |
USB connector |
USB Mini-b connector |
1 |
1.31 |
1.31 |
Crystal |
16 MHz Oscillator |
1 |
1.00 |
1.00 |
Total Cost: |
39.65 |
The cost of the components required to enable our instrument was under $40, as shown in Table 1.
Table 1. Bill of materials
- The color number in Figure 1B is unclear. I think it is better to share the PCB layout file and make it open source to gain early adopter interest and improve the current prototype faster in the future.
Answer: Currently we don’t have any plan to publish it open source. According to your suggestion, the figure numbers are edited as follows;
Figure 1. (A) Schematics of the hardware. STM32F373: Microcontroller, ARM M4 cortex architecture, Built-in ADC, DAC and USB; MAX4617: 8:1 Analog Multiplexer with low current leakage; A1: Operational amplifier with a gain of 1.5; A2: Control amplifier to drive counter electrode; A3: Buffer for reference electrode; TIA: Transimpedance amplifier; A4: ADC driver with 2/3 gain; REF5045: 4.5V reference voltage; REF5030: 3.0V Reference voltage to power analog peripherals of MCU; MCP1703: 3.3V Regulator to power digital peripherals of MCU; CE: Counter electrode; RE: Reference electrode; WE: Working electrode. (B) Printed circuit board. 1: USB connector for communication; 2: 3.3V Regulator; 3:4.5 V reference voltage; 4: 3.0 V reference voltage; 5: ADC Driver; 6: Op-amp; 7: Op-amp; 8: Op-amp; 9: USB connector for electrodes; 10: Analog switch; 11: Microcontroller
- How about the user interface of this system? Is this using a mobile app or PC/laptop? Please clearly explain and add the flow chart on how to use the prototype. The app also can be shared as an open source.
Answer: According to your comment, the section below is added to the manuscript.
User interface was written in C++ by using QT framework for desktop computers and designed by GalvanoPlot (Ä°zmir, Turkey). A virtual COM port was created for USB communication between computer and MCU. An example experiment window for CV is shown in Figure 2. For CV experiments, users enter potential scan range, scan rate, and current resolution. For DPV experiments, users enter scan range, step voltage, pulse period, pulse width, pulse amplitude, and current resolution.
Figure 2. Screenshot of the user interface designed in Qt Framework.
- It is better to show the photo of the prototype during the experimental setup to gain readers' interest in the real size of the device.
Answer: According to your comment, the experimental setup is added to the experimental section as Figure 3 as given below.
“The experimental setup containing electrochemical cell and three electrode system in combination with the developed instrument is given in Figure 3.”
Figure 3. Experimental setup.
- Regarding CV performance, why are the peaks of the prototype higher than the commercial? How much is the difference (in %) of peaks? Please add CV response from the commercial potentiostat.
Answer: According to your comment, the corresponding anodic current (Ia) and cathodic current (Ic) and the electroactive surface area of PGE measured and calculated by commercial or developed potentiostat (this work) are listed in Table 2 and given in Results and discussion section. According to the CV results, the measured anodic and cathodic peak currents (Ia , Ic) with both instrument are given in Table Y. With the developed potentiostat, Ia and Ic were 16.6 % and 35.1 % higher than the one measured with commercial potentiostat, respectively. These results are added to the manuscript as below;
Another experiment was performed with both devices in order to calculate the electroactive surface area (A) of PGE that was calculated using the Randles–Sevcik equation [24]. The corresponding anodic current (Ia) and cathodic current (Ic) and the electroactive surface area of PGE was listed in Table 2. With the developed potentiostat, Ia and Ic were 16.6 % and 35.1 % higher than the one measured with commercial potentiostat, respectively.
Table 2. The average values (n=3) of anodic and cathodic peak currents (Ia , Ic), and electroactive surface areas (A) for PGE measured and calculated by commercial or developed potentiostat (this work).
Device |
Ia (µA) |
Ic (µA) |
A (cm2) |
Commercial potentiostat |
96.44 ± 1.22 |
98.57 ± 0.28 |
0.291 |
Developed potentiostat |
112.44 ± 1.08 |
133.17 ± 3.63 |
0.339 |
- Figure 4 needs to resize. It is unclear and too small.
Answer: Figure 4 (new Figure 6) is resized according to your comment
- Could you check the DPV performance of K3Fe(CN)6 and K4Fe(CN)6 solution on a varied concentration? I want to know the signal responses because it seems the DPV peaks are not sharp in the data in Figure 4A-D.
Answer: As we mentioned in experimental past of manuscript, a digital low-pass filter with a 5 Hz cut-off frequency was applied to all measure-ments by the developed device. Since, no post-filtering was applied to the experiments with Autolab PGSTAT-302, raw data was recorded for each measurement. According to your comment, another experiment was performed and consequently the oxidation signal of [Fe(CN)6]3-/[Fe(CN)6]4- (1: 1) redox couple in various concentrations was measured with commercial and developed potentiostat (this work) via DPV. The voltammograms and the table related to the measured oxidation signal values are given below.
Figure. DPVs representing the oxidation signal of various concentrations (in 1: 1 molar ratio) of the [Fe(CN)6]3-/[Fe(CN)6]4- redox couple measured with (A) commercial instrument (B) developed potentiostat (this work).
Table. The average oxidation signal (n=3) of various concentrations (in 1: 1 molar ratio) of the [Fe(CN)6]3-/[Fe(CN)6]4- redox couple measured with commercial or developed potentiostat (this work) via DPV.
[Fe(CN)6]3-/4- Concentration (mM) |
Oxidation signal measured with/ |
|
Commercial potentiostat |
Developed potentiostat |
|
1 |
108.82 ± 3.19 µA |
87.8 ± 3.32 µA |
2 |
146.61 ± 11.95 µA |
139.11 ± 30.14 µA |
3 |
189.40 ± 29.74 µA |
160.15 ± 6.87 µA |
4 |
225.46 ± 25.26 µA |
221.46 ± 6.14 µA |
5 |
253.48 ± 19.12 µA |
299.72 ± 50.10 µA |
- It seems the signal smoothness on the prototype still quite rough. How to improve the signal? Add the suggestions.
Answer: According to your comment, our suggestions are added to the manuscript as below:
“Number of measurement points effect the signal smoothness. If high number of samples are taken by decreasing the step size, smoother voltammograms can be obtained. But in that case, a digital low-pass filter should be used to filter interferences such as 50/60 Hz. On the other hand, if the step size is big, which means period is longer as well, then the signal can be averaged for at least 20 ms to filter power line interferences without using more complex software techniques.”
- Add a comparison table for the performance of the current prototype with several commercial products and other previous research
Answer: According to your comment, a comparison table for the performance of developed potentiostat and commercial products or other prototypes are added to the conclusion and given below.
Table 4. The comparison of the performance of the developed potentiostat and the other commercial ones.
Device |
Electrochemical Method |
Electrode type |
Analyte |
LOD |
Reference |
LMP91000 |
CV |
Gold IDE |
Cortisol |
1 pM |
[15] |
LMP91000 |
CV, Amperometry |
Carbon SPE |
Cortisol |
74.0 nmol L-1 |
[32] |
ADucM355 |
SWV |
Glassy Carbon Electrode |
TNT |
0.25 mg/L |
[33] |
Custom AFE |
Chronoamperometry |
Carbon SPE |
HRP |
0.83 ng mL-1 |
[34] |
Custom AFE |
CV, DPV |
Carbon SPE |
Cd2+, Pb2+ |
1 µg L-1 Cd2+ 0.5 µg L-1 Pb2+ |
[20] |
Autolab PGSTAT-302 |
CV, DPV |
PGE |
ssDNA, dsDNA |
1.65 µg/mL ssDNA, 0.96 µg/mL dsDNA |
This work |
Current prototype |
CV, DPV |
PGE |
ssDNA, dsDNA |
1.25 µg/mL ssDNA, 1.10 µg/mL dsDNA |
This work |

Reviewer 2 Report
The manuscript is focused on the detection of nucleic acids and biomolecular interactions using a miniaturized hand-held USB powered instrument. The topic is of high practical interest and has clear social impact on human well-being.
The manuscript is well written and has a high readability. The experimental section fully covers all aspects making easy understandable data obtained. The advantage of this work is the comparison with standard equipment widely used in electroanalytical laboratories all over the world. The results are logically explained and conclusions are fully supported with experimental data.
I have just two minor remarks:
1. Figure 4 is almost unreadable and has to be presented in a larger size (probably just two plots in one raw).
2. Tables 1 and 2 can be arrange to one by addition another one column "Type of DNA" which will be the first one and contain two lines (ssDNA and dsDNA) consisting each one of four merged lines corresponding to interaction time.
Author Response
Micromachines
micromachines-1890500
Title:
A low-cost, high-resolution potentiostat for electrochemical detection of nucleic acids and biomolecular interactions
12th September 2022
The list of answers to the comments of reviewers
Thank you for valuable comments of Reviewer#1, Reviewer#2, and Reviewer#3. Manuscript is revised/corrected according to each comment pointed by reviewers. The revised/corrected parts in the manuscript are shown highlighted in yellow.
Reviewers' comments:
Reviewer 2
Comments and Suggestions for Authors
The manuscript is focused on the detection of nucleic acids and biomolecular interactions using a miniaturized hand-held USB powered instrument. The topic is of high practical interest and has clear social impact on human well-being.
The manuscript is well written and has a high readability. The experimental section fully covers all aspects making easy understandable data obtained. The advantage of this work is the comparison with standard equipment widely used in electroanalytical laboratories all over the world. The results are logically explained and conclusions are fully supported with experimental data. I have just two minor remarks:
- Figure 4 is almost unreadable and has to be presented in a larger size (probably just two plots in one raw).
Answer: Authors would like to thank to the Reviewer #2 for their valuable comments. Figure 4 (new Figure 6) is resized according to your comment.
- Tables 1 and 2 can be arrange to one by addition another one column "Type of DNA" which will be the first one and contain two lines (ssDNA and dsDNA) consisting each one of four merged lines corresponding to interaction time.
Answer: According to your suggestion, Table 1 and Table 2 is arranged to one table and given as Table 3 in the main text as below.
Table 3. The average oxidation signals of BSA and guanine (n=3) measured before/after interaction (Ibefore , Iafter) performed in different times with the decrease ratios obtained at both signals calculated after interaction of BSA and ssDNA or dsDNA in different interaction times.
Type of DNA |
Time (min) |
BSA Ibefore (µA) |
BSA Iafter (µA) |
Decrease % at BSA signal |
Guanine Ibefore (µA) |
Guanine Iafter (µA) |
Decrease % at guanine signal |
ssDNA |
5 |
1.19 ± 0.13 |
0.50 ± 0.12 |
58 % |
4.11 ± 0.44 |
2.91 ± 0.54 |
29 % |
15 |
1.51 ± 0.03 |
0.88 ± 0.09 |
41 % |
1.99 ± 0.56 |
51 % |
||
30 |
2.17 ± 0.20 |
0.57 ± 0.09 |
73 % |
2.12 ± 0.26 |
48 % |
||
60
|
2.36 ± 0.11
|
1.14 ± 0.17 |
52 % |
1.97 ± 0.03 |
52 % |
||
dsDNA
|
5 |
1.19 ± 0.13 |
0.98 ± 0.07 |
17 % |
2.27 ± 0.25 |
1.28 ± 0.08 |
43 % |
15 |
1.51 ± 0.03 |
1.04 ± 0.08 |
31 % |
0.93 ± 0.22 |
38 % |
||
30 |
2.17 ± 0.20 |
1.24 ± 0.07 |
42 % |
0.77 ± 0.04 |
66 % |
||
60 |
2.36 ± 0.11 |
1.18 ± 0.04 |
69 % |
0.70 ± 0.16 |
69 % |

Reviewer 3 Report
Erdem and coworkers reported the development of a hand-held, USB powered miniaturized instrument for electrochemical detection of DNAs and protein-DNA interactions. The experimental design was straightforward, and the observations were carefully documented. Valuably, the authors compared the performance of their sensor to a widely used electrochemical analyzer under the same condition and showed reliable performance. Only few minor points need to be addressed before publishing on Micromachines.
1. The method of LOD calculation should be included in the main text or method section.
2. Figures 4E-H are not introduced in the main text. The conclusion from these panels should be provided.
3. BSA has unspecific binding to DNAs, which may not be a good representation of a specific protein-DNA interaction event. Nevertheless, this data is still valuable to show the sensing system is sensitive enough to detect unspecific interactions. It is suggested to include this point in the discussion.
Author Response
Micromachines
micromachines-1890500
Title:
A low-cost, high-resolution potentiostat for electrochemical detection of nucleic acids and biomolecular interactions
12th September 2022
The list of answers to the comments of reviewers
Thank you for valuable comments of Reviewer#1, Reviewer#2, and Reviewer#3. Manuscript is revised/corrected according to each comment pointed by reviewers. The revised/corrected parts in the manuscript are shown highlighted in yellow.
Reviewers' comments:
Reviewer 3
Comments and Suggestions for Authors
Erdem and coworkers reported the development of a hand-held, USB powered miniaturized instrument for electrochemical detection of DNAs and protein-DNA interactions. The experimental design was straightforward, and the observations were carefully documented. Valuably, the authors compared the performance of their sensor to a widely used electrochemical analyzer under the same condition and showed reliable performance. Only few minor points need to be addressed before publishing on Micromachines.
- The method of LOD calculation should be included in the main text or method section.
Answer: Authors would like to thank to the Reviewer #3 for their valuable comments. According to your suggestion, the method of LOD calculation is added to the main text as below;
“The limit of detection (LOD) was estimated as explained by Miller and Miller [28] with a regression equation and the definition as “y = yB + 3SB” (yB is the signal of blank solution and SB is the Standard deviation of the blank solution). The values of LOD obtained by the developed instrument and Autolab PGSTAT-302 are calculated as 1.25 µg/mL and 1.65 µg/mL, respectively.”
- Figures 4E-H are not introduced in the main text. The conclusion from these panels should be provided.
Answer: According to your suggestion, the missing conclusion related to the some graphs of Figures 4 (new Figure 6) is added to the main text as below,
“The line graphs (Figure 6B-D-F-H) representing the average guanine oxidation signals related to the ssDNA or dsDNA immobilized PGE in case of different DNA concentrations to see the relationship between DNA concentration and current. The current is proportionally increased in case of increased DNA concentration until the PGE surface is saturated.”
- BSA has unspecific binding to DNAs, which may not be a good representation of a specific protein-DNA interaction event. Nevertheless, this data is still valuable to show the sensing system is sensitive enough to detect unspecific interactions. It is suggested to include this point in the discussion.
Answer:
As we emphasize in our manuscript, it was aimed in our study to apply our miniature device for electrochemical monitoring the surface-confined interaction of proteins with DNA. Albumin, as the most abundant and the major protein of the blood, plays an important role in many biological processes. In fact, human serum albumin can bind several molecules such as bases, nucleotides, RNA and DNA [23]. Under this aim, the target protein was chosen as BSA in the present study. So that the effect of its interaction with DNA upon to electrode response was detected electrochemically by using DNA modified electrode in combination with both device, commercial potentiostat and developed one.
According to your comment, the required discussion is added to the conclusion as below,
“Since BSA can bind several molecules including DNA, we aimed to apply our miniature device for electrochemical monitoring the surface-confined interaction of DNA with BSA. Our results can be considered as beneficial to present that the sensing system is sensitive enough to detect surface-confined interaction in contrast to the results obtained by commercial one.”

Round 2
Reviewer 1 Report
The data required for the revision have been provided. Therefore, the manuscript is adequate to be accepted.